# Characteristics of individuals who received a complete, 2-dose mpox vaccine regimen as part of the public health response to the mpox epidemic in Ontario, Canada

Ramandip Grewal[1,2,3]*, Cindy Lau[3], Jeffrey C. Kwong[1,2,3,4,5,6,7], Ann N. Burchell[2,3,5,8], Lindsay Friedman[4], Christine Navarro[1,2,4], Evaezi Okpokoro[9], Darrell H. S. Tan[6,8,10,11,12], Austin Zygmunt[4,13], Li Bai[3], Sharmistha Mishra[2,3,6,8,10,11,12‡], Sarah A. Buchan[1,2,3,4,11‡], for the Canadian Immunization Research Network (CIRN) Investigators

1 Centre for Vaccine Preventable Diseases, University of Toronto, Toronto, Ontario, Canada, 2 Dalla Lana School of Public Health, University of Toronto, Toronto, Ontario, Canada, 3 Populations and Public Health, ICES, Toronto, Ontario, Canada, 4 Communicable Disease Control, Public Health Ontario, Toronto, Ontario, Canada, 5 Department of Family and Community Medicine, Temerty Faculty of Medicine, University of Toronto, Toronto, Ontario, Canada, 6 Institute of Medical Science, Temerty Faculty of Medicine, University of Toronto, Toronto, Ontario, Canada, 7 University Health Network, Toronto, Ontario, Canada, 8 MAP Centre for Urban Health Solutions, Li Ka Shing Knowledge Institute, Unity Health Toronto, Toronto, Ontario, Canada, 9 International Research Centre of Excellence, Institute of Human Virology Nigeria, Abuja, Nigeria, 10 Department of Medicine, Temerty Faculty of Medicine, University of Toronto, Toronto, Ontario, Canada, 11 Institute of Health Policy, Management and Evaluation, University of Toronto, Toronto, Ontario, Canada, 12 Division of Infectious Diseases, St. Michael's Hospital, Toronto, Ontario, Canada, 13 Department of Family Medicine, University of Ottawa, Ottawa, Ontario, Canada

‡ These authors are co-senior authors on this work.
* r.grewal@utoronto.ca

## Abstract

In May 2022, an outbreak of mpox emerged in Canada. In June 2022, the province of Ontario began offering first doses of a 2-dose regimen of Modified Vaccinia Ankara–Bavaria Nordic (MVA-BN) to those at high risk of exposure. Second doses became available in September 2022. To help increase dose 2 access and uptake, we sought to understand how individuals who received 2 doses differed from those who received only 1 dose. We conducted a cross-sectional study using population-level data among individuals who received ≥1 dose of MVA-BN between June 6, 2022 and October 31, 2023 in Ontario. We used age-adjusted Poisson regression to examine the association between demographic, social, and economic characteristics; co-morbidities; and proxies for sexual exposure (e.g., bacterial sexually transmitted infection [STI] diagnoses) and proxies for healthcare engagement (e.g., syphilis test-ing, past receipt of other vaccines) with MVA-BN dose 2 receipt. Among 33,012 individuals with ≥1 MVA-BN dose, 38.2% (12,620) received 2 doses. Receipt of dose 2 versus only dose 1 was associated with region (e.g., higher in Ottawa compared with Toronto [prevalence ratio, PR = 1.08, 95% confidence interval, CI 1.06-1.09]); syphilis testing (≥4 tests PR = 1.12, 95%CI 1.11-1.14) or receiving a COVID-19, influenza, or

which permits unrestricted use, distribution, and reproduction in any medium, provided the original author and source are credited.

**Data availability statement:** The dataset from this study is held securely in coded form at ICES. While legal data sharing agreements between ICES and data providers (e.g., health-care organizations and government) prohibit ICES from making the dataset publicly available, access may be granted to those who meet pre-specified criteria for confidential access, available at www.ices.on.ca/DAS (email: das@ices.on.ca). The full dataset creation plan and underlying analytic code are available from the authors upon request, understanding that the computer programs may rely upon coding tem-plates or macros that are unique to ICES and are therefore either inaccessible or may require modification. Further, our dataset includes data from Public Health Ontario (PHO) who cannot disclose the underlying data. Doing so would compromise individual privacy contrary to PHO's ethical and legal obligations. Restricted access to the data may be available under conditions prescribed by the Ontario Personal Health Information Protection Act, 2004, the Ontario Freedom of Information and Protection of Privacy Act, the Tri-Council Policy Statement: Ethical Conduct for Research Involving Humans (TCPS 2 (2022)), and PHO privacy and ethics policies. Data are available for researchers who meet PHO's criteria for access to confidential data. Information about PHO's data access request process is available on-line at https://www.publichealthontario.ca/en/data-and-analysis/using-data/data-requests (data@oahpp.ca).

**Funding:** This study was supported by a grant from the Public Health Agency of Canada and the Canadian Institutes of Health Research (CNF 151944, to JCK) through the Canadian Immunization Research Network. The Canada-Africa Mpox Partnership, of which this study is one component, was also supported by the Canadian Institutes of Health Research Rapid Mpox Research (MRR-184812 to DHST). This study was also supported by ICES, which is funded by an annual grant from the Ontario Ministry of Health and the Ministry of Long-Term Care. JCK is supported by a clinician-scientist award and ANB is supported by a non-clinician scientist award from the University of Toronto Department of Family and Community Medicine. SM, DHST, and ANB are supported by Tier 2 Canada Research Chairs. The funders had no role in study design, data

other vaccine (PR = 1.12, 95%CI 1.11-1.14) in the year before dose 1; and syphilis testing (≥4 tests PR = 1.19, 95%CI 1.18-1.20) or bacterial STI diagnoses >3 months after dose 1 (≥4 diagnoses PR = 1.07, 95% CI 1.05-1.08). Refugees were less likely to get dose 2 versus Canadian-born individuals or long-term immigrants (PR = 0.93, 95%CI 0.91-0.95).Our findings suggest lower healthcare access and/or engage-ment may play a role in limiting dose 2 receipt in Ontario. Public messaging around availability and eligibility of second doses, tailored strategies for eligible refugees, increased access outside healthcare venues, and adopting promotion strategies from regions with high uptake, may help increase dose 2 coverage.

## Introduction

Mpox is an infectious disease caused by the MPXV virus [1]. In May 2022, outbreaks of Clade IIb mpox emerged in countries where the virus was not previously endemic, including Canada. The virus can be transmitted through close or intimate contact [1]. Mpox disproportionately affected gay, bisexual, and other men who have sex with men (GBM); 96% of cases in Canada from 2022-23 were reported among individuals identifying as GBM [2]. In Canada, as in many other high-income settings across the global north, the Modified Vaccinia Ankara–Bavaria Nordic (MVA-BN) vaccine was deployed for the mpox response. MVA-BN is recommended to be given as a 2-dose regimen at least 28 days apart with studies estimating 2-dose effectiveness ranging between 66–89% [1,3].

In June 2022, Ontario, Canada, the country's most populous province (population ~16 million), [4] began offering publicly funded first doses of MVA-BN as pre-exposure prophylaxis to individuals at high risk of exposure to mpox. This was defined as self-identifying as engaging in sex work and GBM reporting one or more of the fol-lowing: diagnosis of a bacterial sexually transmitted infection (STI) in the previous two months, currently engaging in or anticipating sex with two or more sex partners, attend-ing sex-on-premises venues, engaging in anonymous sex, or being a sexual contact of an individual who engages in sex work [5]. Immunocompromised and pregnant individ-uals were also eligible if they were contacts of people at risk, as defined above. Due to limited supply at the time of the initial roll-out, second doses were temporarily delayed so more individuals at risk of exposure could receive their first dose. As of September 30, 2022, publicly funded second doses were made widely available to the same eligi-ble populations. Nonetheless, as of November 30, 2024, only 38.2% of Ontarians who received 1 dose of MVA-BN had received their second dose [6].

In 2024, there was a sustained increase in mpox Clade IIb cases in Ontario due to ongoing community transmission of MPXV among GBM [7]. Following a period of low case reporting in 2023 with a total of only 33 cases, from January 1 to November 30, 2024, 275 cases were reported [6]. Of these cases, 84% either had only 1 dose or were unvaccinated [6]. There is a pressing need to increase uptake of the full 2-dose regimen of MVA-BN to increase protection against mpox in those at risk of exposure. To help increase dose 2 access and uptake, we sought to understand characteristics

collection and analysis, decision to publish, or preparation of the manuscript. The opinions, results, and conclusions reported in this paper are those of the authors and are independent from the funding sources.

**Competing interests:** SM, CL, JCK, ANB, LF, CN, EO, AZ, LB, SM, SAB: The authors have declared that no competing interests exist. RG: I have read the journal's policy and the authors of this manuscript have the following competing interests. "The Centre for Vaccine Preventable Diseases, where RG is an Assistant Professor, is supported by the Dalla Lana School of Public Health, which funds infrastructure, and faculty and staff salaries through a mix of funding sources, including donations from vaccine manufacturers. We have a robust set of governance processes at the school to ensure independent operation of the Centre and its Faculty and Staff. RG holds the Sanofi Early Career Investigator in Vaccine Access Research at the University of Toronto." DHST: I have read the journal's policy and the authors of this manuscript have the following competing interests. "DHST's institution has received support from Gilead Sciences for investigator-initiated research and from Glaxo Smith Kline for participation in industry-sponsored clinical trials." The other authors declare no competing interests.

of individuals who received 2 doses of MVA-BN as compared with only 1 dose in Ontario, Canada.

## Methods

### Ethics statement

Ethics approval was obtained from Public Health Ontario's Ethics Review Board. The data were accessed on October 31, 2023 (the last date of the data extraction) for this study. ICES is a prescribed entity under Ontario's Personal Health Information Protection Act (PHIPA). Section 45 of PHIPA authorises ICES to collect personal health information, without consent, for the purpose of analysis or compiling statistical information with respect to the management of, evaluation or monitoring of, the allocation of resources to or planning for all or part of the health system. Projects that use data collected by ICES under section 45 of PHIPA, and use no other data, are exempt from REB review. The use of the data in this project is authorised under section 45 and approved by ICES' Privacy and Legal Office.

### Study design, setting, and population

We used an observational, cross-sectional study using retrospective cohort data [8]. The secondary data come from linked public health and health administrative databases from the province of Ontario, Canada. The province has a single payer healthcare system, which means that there is a single repository for health data on all individuals who are residents of Ontario and who receive health care through the single payer system. Thus, the health administrative databases include person-level data on vaccination, laboratory tests, as well as neighborhood-level measures [9]. The details of the health administrative data sources are provided in the Supplementary Methods in S1 Text.

Ontario residents who received 1 dose of MVA-BN between June 6, 2022 and October 31, 2023 were included. We excluded non-Ontario residents, those without provincial health insurance, those who hadn't accessed care in the province in the previous 8 years, those with a missing birthdate, and residents of long-term care. We also excluded 1) individuals with a known positive mpox test before or after their first dose (n = 221) since vaccination was not recommended for individuals who had already been infected and 2) those who had a first dose ≤28 days prior to the study end date of October 31, 2023 (n = 64) as they would not yet be eligible for dose 2. All individuals were followed until the end of the study period.

### Data sources

We linked laboratory, MVA-BN vaccination (all doses administered were captured in a public health centralized database), reportable diseases, and health administrative datasets using unique encoded identifiers (i.e., data were deidentified prior to analysis) and analyzed them at ICES.

**Outcome and characteristics of interest**

Our outcome of interest was receipt of a second dose of MVA-BN. Characteristics of interest were drawn from available data across various databases (S1 Text), and included: demographic characteristics (i.e., age, sex, and geographic region), social and economic factors (i.e., immigration status, and neighbourhood-level factors including income and self-identified visible minority indices), co-morbidities that may influence vaccine uptake (i.e., immunocompromised status and history of HIV), proxies for sexual exposure (i.e., bacterial STI diagnoses [syphilis, chlamydia, or gonorrhea] and use of HIV pre-exposure prophylaxis [PrEP] as defined by a previously developed algorithm [10]) and proxies for healthcare engagement (receipt of other vaccines and number of physician office visits) (S1 Text, S1 Table, S1 Figure, S2 Figure,). Syphilis testing was used as a proxy for both sexual exposure and healthcare engagement. We looked at syphilis testing and bacterial STI diagnoses before dose 1 receipt as a proxy for sexual exposure prior to vaccination and >3 months after receiving dose 1 to determine potential ongoing sexual exposure after initiating vaccination.

**Analysis**

We described the study population using proportions and medians and compared individuals who received only 1 dose of MVA-BN to those who received 2 doses using standardized differences, chi-square tests for categorical variables, and the Kruskal Wallis test for continuous variables (following tests of normality). We examined the distribution of days between doses among those who received their first dose before September 30, 2022, and separately for those who received their first dose on or after September 30, 2022 (i.e., after second doses were made available). We used Poisson regression with robust standard errors adjusted for age to produce prevalence ratios (PR) with 95% confidence intervals (CI) to examine the association between the characteristics of interest and MVA-BN dose 2 receipt among dose 1 recipients. We examined the CI to discern associations. In accordance with our objective of a descriptive study, age-adjustment was done to age-standardize our description of associations and not to conduct predictive modeling or causal inference [11]. We examined the CI to discern programmatically meaningful differences. We used SAS version 9.1 (SAS Institute Inc., Cary, NC) for all analyses.

**Results**

After exclusions, 33,012 Ontario residents who had received at least 1 dose of the MVA-BN vaccine between June 6, 2022, and October 31, 2023, were included in this study (S2 Fig). Of these individuals, 61.8% (N = 20,392) received only 1 dose and 38.2% (N = 12,620) received 2 doses. Table 1 summarizes the participant characteristics by receipt of first and second doses, and S2 Table providing stratifications with row percentages. Among all those who received 2 doses, the median number of days between doses was 125 days (interquartile range [IQR] 97–230) (Table 1). Among individuals who received their first dose on or after September 30, 2022, more received their second dose and the interval between doses was generally shorter than those who received their first dose before September 30 (median 42 days [IQR 32–70] versus 135 days [IQR 108–242]) (Table 1; S3 Fig). Compared to individuals who received only 1 dose, individuals who received a second dose were older (median age 40 years [IQR 33–53] versus 37 years [IQR 30–51]). More second dose versus only dose 1 recipients were male (96.6% versus 91.9%), and in the year prior to their first dose of MVA-BN, more received COVID-19, influenza, or another vaccine (93.6% versus 87.5%) or were tested for syphilis (54.0% vs. 44.0%). A higher proportion of 2 dose versus only 1 dose recipients were from Ottawa whereas a lower proportion of 2 dose recipients were from Peel/York/Durham/Halton.

Males versus females (PR = 1.14, 95%CI 1.12-1.16), residents of the city of Ottawa versus Toronto (PR = 1.08, 95%CI 1.06-1.09), individuals who received another vaccine in the year prior to their first dose (PR = 1.12, 95%CI 1.11-1.14) versus those who did not, and those with a syphilis test in the year prior to their first dose were more likely to have received a second dose of MVA-BN (Table 2). The likelihood of receiving a second dose increased as the number of syphilis tests

**Table 1. Descriptive characteristics of individuals who received at least 1 dose of the MVA-BN vaccine in Ontario, Canada, between June 6, 2022, to October 31, 2023 and comparing characteristics of those who received dose 2 versus those who received only dose 1.**

| Characteristics | At least 1 dose, N(%) | 1 dose only, N(%) | 2 doses, N(%) | SD | P-value[c] |
|---|---|---|---|---|---|
| | N = 33,012 | N = 20,392 | N = 12,620 | | |
| Number of days between 1st and 2nd dose of MVA-BN | | | | | |
| Median (IQR) | – | – | 125 (97-230) | – | – |
| First dose before or on/after September 30, 2023 | | | | | |
| Before September 30 | 29,650 (89.8%) | 18,565 (91.0%) | 11,085 (87.8%) | 0.10 | <.001 |
| | | | | | – |
| Median number of days between doses (IQR) | – | – | 135 (108-242) | | – |
| On/After September 30 | 3,362 (10.2%) | 1,827 (9.0%) | 1,535 (12.2%) | 0.10 | <.001 |
| Median number of days between doses (IQR) | – | – | 42 (32-70) | | – |
| Age at dose 1 | | | | | |
| Median (IQR) | 38 (31-52) | 37 (30-51) | 40 (33-53) | 0.22 | <.001 |
| Age group (years) at dose 1 | | | | | |
| 0-17 | 34 (0.1%) | 30-34 (0.1-0.2%)[a] | ≤5 (0.0%)[a] | 0.04 | <.001 |
| 18-24 | 2,290 (6.9%) | 1,767–1,771 (8.7%)[a] | 520-524 (4.1-4.2%)[a] | 0.19 | |
| 25-29 | 4,519 (13.7%) | 3,174 (15.6%) | 1,345 (10.7%) | 0.15 | |
| 30-39 | 10,754 (32.6%) | 6,633 (32.5%) | 4,121 (32.7%) | 0.00 | |
| 40-49 | 6,078 (18.4%) | 3,341 (16.4%) | 2,737 (21.7%) | 0.14 | |
| 50-59 | 5,398 (16.4%) | 3,105 (15.2%) | 2,293 (18.2%) | 0.08 | |
| ≥60 | 3,939 (11.9%) | 2,340 (11.5%) | 1,599 (12.7%) | 0.04 | |
| Sex | | | | | |
| Female | 2,089 (6.3%) | 1,661 (8.1%) | 428 (3.4%) | 0.20 | <.001 |
| Male | 30,923 (93.7%) | 18,731 (91.9%) | 12,192 (96.6%) | 0.20 | |
| Reason for immunization | | | | | |
| Post-exposure | 218 (0.7%) | 192 (0.9%) | 26 (0.2%) | 0.10 | <.001 |
| Pre-exposure | 32,794 (99.3%) | 20,200 (99.1%) | 12,594 (99.8%) | 0.10 | |
| Geographic region | | | | | |
| Toronto | 20,420 (61.9%) | 12,730 (62.4%) | 7,690 (60.9%) | 0.03 | <.001 |
| Ottawa | 3,689 (11.2%) | 1,927 (9.4%) | 1,762 (14.0%) | 0.14 | |
| Peel, York, Durham, Halton | 3,467 (10.5%) | 2,519 (12.4%) | 948 (7.5%) | 0.16 | |
| Hamilton, Niagara, London, Windsor | 2,107 (6.4%) | 1,393 (6.8%) | 714 (5.7%) | 0.05 | |
| Rest of Ontario | 3,329 (10.1%) | 1,823 (8.9%) | 1,506 (11.9%) | 0.10 | |
| Neighbourhood income quintile | | | | | |
| Missing | 103 (0.3%) | 68 (0.3%) | 35 (0.3%) | 0.01 | 0.044 |
| 1 (lowest) | 7,940 (24.1%) | 4,999 (24.5%) | 2,941 (23.3%) | 0.03 | |
| 2 | 7,363 (22.3%) | 4,501 (22.1%) | 2,862 (22.7%) | 0.01 | |
| 3 | 6,209 (18.8%) | 3,767 (18.5%) | 2,442 (19.4%) | 0.02 | |
| 4 | 5,392 (16.3%) | 3,368 (16.5%) | 2,024 (16.0%) | 0.01 | |
| 5 (highest) | 6,005 (18.2%) | 3,689 (18.1%) | 2,316 (18.4%) | 0.01 | |
| Neighbourhood visible minorities quintile | | | | | |
| Missing | 2,195 (6.6%) | 1,378 (6.8%) | 817 (6.5%) | 0.01 | <.001 |
| 1 (lowest) | 1,226 (3.7%) | 690 (3.4%) | 536 (4.2%) | 0.05 | |
| 2 | 2,170 (6.6%) | 1,276 (6.3%) | 894 (7.1%) | 0.03 | |
| 3 | 5,576 (16.9%) | 3,372 (16.5%) | 2,204 (17.5%) | 0.02 | |
| 4 | 12,618 (38.2%) | 7,750 (38.0%) | 4,868 (38.6%) | 0.01 | |

*(Continued)*

| Characteristics | At least 1 dose, N(%) | 1 dose only, N(%) | 2 doses, N(%) | SD | P-value[c] |
|---|---|---|---|---|---|
| | N=33,012 | N=20,392 | N=12,620 | | |
| 5 (highest) | 9,227 (28.0%) | 5,926 (29.1%) | 3,301 (26.2%) | 0.06 | |
| Recent immigration | | | | | |
| Refugees | 1,097 (3.3%) | 782 (3.8%) | 315 (2.5%) | 0.08 | <.001 |
| <5 years ago | 1,988 (6.0%) | 1,235 (6.1%) | 753 (6.0%) | 0.00 | |
| 5-10 years ago | 895 (2.7%) | 562 (2.8%) | 333 (2.6%) | 0.01 | |
| >10 years ago (but after1985) | 3,662 (11.1%) | 2,322 (11.4%) | 1,340 (10.6%) | 0.02 | |
| Born in Canada or immigrated before 1985 | 25,370 (76.9%) | 15,491 (76.0%) | 9,879 (78.3%) | 0.06 | |
| Received any vaccine before dose 1, in past year (COVID-19, influenza, or other) | | | | | |
| No | 3,362 (10.2%) | 2,554 (12.5%) | 808 (6.4%) | 0.21 | <.001 |
| Yes | 29,650 (89.8%) | 17,838 (87.5%) | 11,812 (93.6%) | 0.21 | |
| Syphilis screening tests before dose 1, past 1 year | | | | | |
| No | 17,231 (52.2%) | 11,427 (56.0%) | 5,804 (46.0%) | 0.20 | <.001 |
| Yes | 15,781 (47.8%) | 8,965 (44.0%) | 6,816 (54.0%) | 0.20 | |
| Number of syphilis screening tests before dose 1, past 1 year | | | | | |
| 0 | 17,231 (52.2%) | 11,427 (56.0%) | 5,804 (46.0%) | 0.20 | <.001 |
| 1 | 5,986 (18.1%) | 3,640 (17.9%) | 2,346 (18.6%) | 0.02 | |
| 2 | 3,734 (11.3%) | 2,133 (10.5%) | 1,601 (12.7%) | 0.07 | |
| 3 | 2,743 (8.3%) | 1,512 (7.4%) | 1,231 (9.8%) | 0.08 | |
| ≥4 | 3,318 (10.1%) | 1,680 (8.2%) | 1,638 (13.0%) | 0.15 | |
| Number of syphilis screening tests >3 months[b] after dose 1 | | | | | |
| 0 | 15,584 (47.2%) | 10,715 (52.5%) | 4,869 (38.6%) | 0.28 | <.001 |
| 1 | 5,508 (16.7%) | 3,490 (17.1%) | 2,018 (16.0%) | 0.03 | |
| 2 | 3,643 (11.0%) | 2,112 (10.4%) | 1,531 (12.1%) | 0.06 | |
| 3 | 3,056 (9.3%) | 1,696 (8.3%) | 1,360 (10.8%) | 0.08 | |
| ≥4 | 5,221 (15.8%) | 2,379 (11.7%) | 2,842 (22.5%) | 0.29 | |
| Monthly rate of syphilis tests after dose 1 up until second dose, death, or Oct 31, 2023 | | | | | |
| Median (IQR) | 0 (0-0) | 0 (0-0) | 0 (0-0) | 0.34 | <.001 |
| Number of bacterial STIs before dose 1, past 3 years | | | | | |
| 0 | 26,155 (79.2%) | 16,285 (79.9%) | 9,870 (78.2%) | 0.04 | 0.009 |
| 1 | 2,690 (8.1%) | 1,597 (7.8%) | 1,093 (8.7%) | 0.03 | |
| 2 | 1,236 (3.7%) | 748 (3.7%) | 488 (3.9%) | 0.01 | |
| 3 | 677 (2.1%) | 403 (2.0%) | 274 (2.2%) | 0.01 | |
| ≥4 | 2,254 (6.8%) | 1,359 (6.7%) | 895 (7.1%) | 0.02 | |
| Number of bacterial STIs >3 months[b] after dose 1 | | | | | |
| 0 | 27,051 (81.9%) | 17,069 (83.7%) | 9,982 (79.1%) | 0.12 | <.001 |
| 1 | 2,498 (7.6%) | 1,349 (6.6%) | 1,149 (9.1%) | 0.09 | |
| 2 | 1,199 (3.6%) | 700 (3.4%) | 499 (4.0%) | 0.03 | |
| 3 | 764 (2.3%) | 447 (2.2%) | 317 (2.5%) | 0.02 | |
| ≥4 | 1,500 (4.5%) | 827 (4.1%) | 673 (5.3%) | 0.06 | |
| Monthly rate of any bacterial STI after dose 1 up until second dose, death, or Oct 31, 2023 | | | | | |

*(Continued)*

**Table 1.** (Continued)

| Characteristics | At least 1 dose, N(%) | 1 dose only, N(%) | 2 doses, N(%) | SD | P-value[c] |
|---|---|---|---|---|---|
| | N = 33,012 | N = 20,392 | N = 12,620 | | |
| Median (IQR) | 0 (0-0) | 0 (0-0) | 0 (0-0) | 0.10 | <.001 |
| History of HIV diagnosis before dose 1 | | | | | |
| No | 28,421 (86.1%) | 17,804 (87.3%) | 10,617 (84.1%) | 0.09 | <0.001 |
| Yes | 4,591 (13.9%) | 2,588 (12.7%) | 2,003 (15.9%) | 0.09 | |
| Number of physician office visits before dose 1, in past year | | | | | |
| No visits | 5,012 (15.2%) | 3,344 (16.4%) | 1,668 (13.2%) | 0.09 | <.001 |
| 1-2 visits | 6,274 (19.0%) | 3,964 (19.4%) | 2,310 (18.3%) | 0.03 | |
| 3-4 visits | 5,615 (17.0%) | 3,400 (16.7%) | 2,215 (17.6%) | 0.02 | |
| 5+ visits | 16,111 (48.8%) | 9,684 (47.5%) | 6,427 (50.9%) | 0.07 | |
| Has a primary care physician | | | | | |
| Not rostered | 2,641 (8.0%) | 1,759 (8.6%) | 882 (7.0%) | 0.06 | <.001 |
| Rostered | 23,146 (70.1%) | 13,980 (68.6%) | 9,166 (72.6%) | 0.09 | |
| Virtually Rostered | 7,225 (21.9%) | 4,653 (22.8%) | 2,572 (20.4%) | 0.06 | |
| PrEP prescription before dose 1, in past year | | | | | |
| No | 31,620 (95.8%) | 19,520 (95.7%) | 12,100 (95.9%) | 0.01 | 0.494 |
| Yes | 1,392 (4.2%) | 872 (4.3%) | 520 (4.1%) | 0.01 | |
| PrEP prescription >3 months[a] after dose 1 | | | | | |
| No | 31,350 (95.0%) | 19,429 (95.3%) | 11,921 (94.5%) | 0.04 | <.001 |
| Yes | 1,662 (5.0%) | 963 (4.7%) | 699 (5.5%) | 0.04 | |
| Moderately or severely immunocompromised (other than HIV) | | | | | |
| No | 30,233 (91.6%) | 18,779 (92.1%) | 11,454 (90.8%) | 0.05 | <.001 |
| Yes | 2,779 (8.4%) | 1,613 (7.9%) | 1,166 (9.2%) | 0.05 | |

IQR = interquartile range. SD = standardized difference. STI = sexually transmitted infection. PrEP = pre-exposure prophylaxis.

[a] Due to institutional privacy policies, any cells ≤5 (except for missing values) must be suppressed and ranges must be provided for complementary cells to prevent back calculation

[b] 3-month lag since healthcare engagement may have increased after dose 1 visit; intent was to assess potential for ongoing exposure to mpox

[c] Chi-squared tests were used to compare proportions. For continuous values, we used non-parametric tests of comparison (Kruskal-Wallis test).

increased (≥4 syphilis tests PR = 1.12, 95%CI 1.11-1.14). Second dose recipients versus first dose only recipients were more likely to have a bacterial STI diagnosis (≥4 diagnoses PR = 1.07, 95% CI 1.05-1.08) or syphilis test (increasing PR with each additional test; ≥4 tests PR = 1.19, 95%CI 1.18-1.20) at least three months after dose 1 receipt. Additionally, individuals receiving MVA-BN immunization for post-exposure versus pre-exposure prophylaxis (PR = 0.81, 95%CI 0.78-0.84), residents of Peel/York/Durham/Halton regions versus Toronto (PR = 0.93, 95%CI 0.92-0.95), and refugees versus individuals born in Canada or long-term immigrants (PR = 0.93, 95%CI 0.91-0.95) were less likely to receive 2 doses of MVA-BN versus only 1 dose. The results of the crude and age-adjusted prevalence ratios are provided in S3 Table.

## Discussion

Among individuals in Ontario, Canada, who received at least 1 dose of MVA-BN between June 2022 and October 31, 2023, only 38.2% went on to receive a second dose during the study period. Dose 2 receipt was associated with

**Table 2. Factors associated with dose 2 receipt versus only receiving dose 1 of MVA-BN vaccine in Ontario, Canada, between June 6, 2022, to October 31, 2023.**

| Variable | Received 2 doses (N, %) (N = 12,620) | Age-adjusted PR (95% CI) | P-value |
|---|---|---|---|
| Sex | | | |
| Female | 428 (20.44) | Ref | |
| Male | 12192 (39.35) | 1.14 (1.12-1.16) | <.0001 |
| Reason for immunization | | | |
| Pre-exposure | 26 (11.93) | Ref | – |
| Post-exposure | 12594 (38.33) | 0.81 (0.78-0.84) | <.0001 |
| Geographic region | | | |
| Toronto | 7690 (37.59) | Ref | – |
| Rest of Ontario | 1506 (45.08) | 1.06 (1.04-1.07) | <.0001 |
| Hamilton, Niagara, London | 714 (33.89) | 0.97 (0.96-0.99) | 0.0003 |
| Ottawa | 1762 (47.63) | 1.08 (1.06-1.09) | <.0001 |
| Peel, York, Durham, Halton | 948 (27.30) | 0.93 (0.92-0.95) | <.0001 |
| Neighbourhood income quintile[a] | | | |
| 1 (lowest) | 2941 (36.96) | Ref | – |
| 2 | 2862 (38.80) | 1.01 (1.00-1.03) | 0.012 |
| 3 | 2442 (39.27) | 1.02 (1.01-1.03) | 0.004 |
| 4 | 2024 (37.44) | 1.01 (1.00-1.02) | 0.228 |
| 5 (highest) | 2316 (38.51) | 1.01 (1.00-1.02) | 0.105 |
| Neighbourhood visible minorities quintile[a] | | | |
| 1 (lowest) | 536 (43.58) | Ref | – |
| 2 | 894 (41.14) | 0.98 (0.96-1.01) | 0.178 |
| 3 | 2204 (39.49) | 0.97 (0.95-0.99) | 0.011 |
| 4 | 4868 (38.50) | 0.97 (0.95-0.99) | 0.002 |
| 5 (highest) | 3301 (35.69) | 0.95 (0.93-0.97) | <.0001 |
| Recent immigration | | | |
| Born in Canada or immigrated before 1985 | 9879 (38.87) | Ref | – |
| Refugees | 315 (28.66) | 0.93 (0.91-0.95) | <.0001 |
| <5 years ago | 753 (37.73) | 1.01 (1.00-1.03) | 0.120 |
| 5-10 years ago | 333 (37.08) | 1.00 (0.98-1.02) | 0.881 |
| >10 years ago (but after1985) | 1340 (36.54) | 0.98 (0.97-0.99) | 0.006 |
| Received any vaccine before dose 1, in past year (COVID-19, influenza, or other) | | | |
| No | 657 (22.63) | Ref | – |
| Yes | 11963 (39.65) | 1.12 (1.11-1.14) | <.0001 |
| Number of syphilis screening tests before dose 1, in past year | | | |
| 0 | 5804 (33.62) | Ref | – |
| 1 | 2346 (39.08) | 1.04 (1.03-1.05) | <.0001 |
| 2 | 1601 (42.84) | 1.07 (1.06-1.08) | <.0001 |
| 3 | 1231 (44.80) | 1.09 (1.07-1.10) | <.0001 |
| ≥4 | 1638 (49.29) | 1.12 (1.11-1.14) | <.0001 |
| Number of syphilis screening tests >3 months[a] after dose 1 | | | |
| 0 | 4869 (31.12) | Ref | – |
| 1 | 2018 (36.64) | 1.04 (1.03-1.06) | <.0001 |

*(Continued)*

Table 2. (Continued)

| Variable | Received 2 doses (N, %) (N=12,620) | Age-adjusted PR (95% CI) | P-value |
|---|---|---|---|
| 2 | 1531 (42.03) | 1.09 (1.07-1.10) | <.0001 |
| 3 | 1360 (44.50) | 1.11 (1.09-1.12) | <.0001 |
| ≥4 | 2842 (54.43) | 1.19 (1.18-1.20) | <.0001 |
| Number of bacterial STIs before dose 1, in past 3 years | | | |
| 0 | 9870 (37.66) | Ref | – |
| 1 | 1093 (40.54) | 1.03 (1.02-1.05) | <.0001 |
| 2 | 488 (39.42) | 1.02 (1.00-1.04) | 0.030 |
| 3 | 274 (40.29) | 1.03 (1.00-1.06) | 0.043 |
| ≥4 | 895 (39.65) | 1.02 (1.00-1.03) | 0.019 |
| Number of bacterial STIs >3 months[a] after dose 1 | | | |
| 0 | 9982 (36.81) | Ref | – |
| 1 | 1149 (46.00) | 1.08 (1.06-1.09) | <.0001 |
| 2 | 499 (41.62) | 1.04 (1.02-1.06) | <.0001 |
| 3 | 317 (41.49) | 1.04 (1.01-1.06) | 0.005 |
| ≥4 | 673 (44.87) | 1.07 (1.05-1.08) | <.0001 |
| History of HIV diagnosis | | | |
| No | 12584 (38.13) | Ref | – |
| Yes | 36 (49.32) | 1.03 (1.02-1.04) | <.0001 |
| Number of physician office visits before dose 1, in past year | | | |
| No visits | 1668 (33.20) | Ref | – |
| 1-2 visits | 2310 (36.73) | 1.02 (1.01-1.03) | 0.002 |
| 3-4 visits | 2215 (39.39) | 1.04 (1.02-1.05) | <.0001 |
| 5+ visits | 6427 (39.82) | 1.04 (1.02-1.05) | <.0001 |
| Has a primary care physician | | | |
| Not rostered | 882 (33.31) | Ref | – |
| Rostered | 9166 (39.54) | 1.03 (1.02-1.04) | <.0001 |
| Virtually Rostered | 2572 (35.50) | 1.01 (1.00-1.03) | 0.081 |
| PrEP prescription before dose 1, in past year | | | |
| No | 12100 (38.19) | Ref | – |
| Yes | 520 (37.33) | 1.00 (0.98-1.02) | 0.730 |
| PrEP prescription >3 months[b] after dose 1 | | | |
| No | 11921 (37.95) | Ref | – |
| Yes | 699 (42.06) | 1.04 (1.02-1.05) | <.0001 |
| Moderately or severely immunocompromised (other than HIV) | | | |
| No | 11454 (37.81) | Ref | – |
| Yes | 1166 (41.93) | 1.01 (0.99-1.02) | 0.418 |

PR = prevalence ratio. IQR = interquartile range. STI = sexually transmitted infection. PrEP = pre-exposure prophylaxis.

[a] Missing values were excluded from analyses.

[b] 3-month lag since healthcare engagement may have increased after dose 1 visit; intent was to assess potential for ongoing exposure to mpox.

demographic and social characteristics, including differences by geographic region and lower 2 dose coverage among individuals identifying as a refugee. Dose 2 receipt was also associated with increased sexual exposure risk and healthcare engagement. For instance, individuals diagnosed with a bacterial STI and tested for syphilis four or more times at

least three months after receiving their first dose were 7% and 19% more likely, respectively, to have received dose 2 versus only dose 1.

The timing of roll out of second doses likely impacted uptake in Ontario. At the time second doses were made available (September 30, 2022), cases of mpox began to decline [12]. Thus, potential reasons for low uptake after second doses were made available may include a decline in media attention, and decreased access to vaccine outside of traditional healthcare venues. MVA-BN vaccination intentions and uptake have been shown to be associated with perceived mpox susceptibility and fear of morbidity [13–15]. Fewer reported cases and a decline in media attention may also have contributed to lowering risk perceptions around mpox, leading to lower prioritization of vaccination. Though delaying second doses was a public health necessity at the time, the shorter wait between doses for those who received dose 1 later in the outbreak may have contributed to higher chances of returning for a second dose. We also saw differences in second dose receipt across Ontario; Ottawa had a higher proportion of the eligible population with a first dose complete the series versus receive only 1 dose compared to Ontario's largest city, Toronto. Differences could be attributed to the strategies used to promote vaccination as well as differences in target population size. Lower coverage of second doses has also been seen across different settings, including other Canadian provinces, the United States, and England [16–18].

Refugees compared to Canadian-born individuals and long-term immigrants were less likely to get dose 2. Newcomers to Canada are known to experience inequities in vaccination, and refugee populations may be particularly vulnerable. [19,20] Refugees may face unique barriers to accessing healthcare services in Canada, such as precarious immigration status, poverty, competing priorities, lack of information, and cultural and language barriers [21–23]. These barriers may have contributed to some individuals not returning for a second dose. Tailored interventions accounting for these factors are needed to reduce inequities in vaccination and other healthcare services in these populations [21].

Our findings suggest lower healthcare access and/or engagement may also play a role in limiting the receipt of second doses and those at higher risk of exposure to mpox were more likely to complete the vaccination series. Individuals who received other vaccines were 12% more likely to get fully vaccinated against mpox. Previous vaccination history as a predictor for vaccine uptake has been seen in other studies on mpox and in GBM populations for other vaccines [14,24,25]. These individuals may be more engaged in their healthcare and could be more accepting of vaccination. We also found an association with syphilis testing before and after dose 1, with an increased likelihood of dose 2 receipt with each additional test, suggesting that increased sexual healthcare engagement may increase chances of completing the series. Clinicians may opportunistically offer mpox vaccination when individuals come in for other sexual healthcare services (i.e., bundling these services). A study in Rhode Island, US, found that bundling human papillomavirus vaccination with HIV testing was found to be feasible and acceptable among young GBM, and led to increased vaccine coverage [26]. Multiple studies have found an association between MVA-BN dose 1 receipt and past STI diagnoses [27–29]. Our findings of higher dose 2 receipt among individuals diagnosed with bacterial STIs after receiving their first dose suggest that individuals potentially at higher risk of ongoing exposure to mpox were more likely to complete their MVA-BN series. It is unknown whether these individuals were seeking vaccination or were offered vaccination during healthcare visits.

Our study is descriptive and focuses only on associations between the presented characteristics and receipt of second doses. As such, these characteristics help provide potentially actionable information for programs around whom to reach to fill gaps, how to reach and tailor services, and where or within what healthcare context. Our study is not designed to explore or examine explanatory or causal pathways nor is it designed as a predictive study. We were also limited in the characteristics that could be explored due to data availability. Nonetheless, our study has several strengths including the use of population-based public health vaccination records linked to administrative data and our large sample size, which allowed us to highlight important factors that may have contributed to MVA-BN series completion and that can inform interventions and policy decisions around MVA-BN vaccination planning.

Our findings suggest that individuals at potentially greater ongoing risk of exposure to mpox may be more likely to receive their second dose of MVA-BN, however, dose 2 coverage remains suboptimal. Nearly two thirds of Ontarians who received their first dose of MVA-BN did not complete the series at the time of this study and receiving 1 dose has been shown to provide only moderate protection against mpox with higher effectiveness seen with 2 doses [3,10]. Additional efforts are needed to ensure those eligible for the mpox vaccine receive both recommended doses. Potential strategies may include better messaging (e.g., around availability, need for, and eligibility of second doses), tailored strategies for eligible refugees, increased access outside healthcare venues, and opportunistically offering vaccination during other healthcare encounters. Moreover, higher vaccine completion in some Ontario regions should encourage learning from and adopting strategies between regions to help increase vaccine uptake locally.

## Supporting information

**S1 Text. Description of linked data repositories.**
(DOCX)

**S1 Fig. Conceptual framework for associations explored between characteristics and MVA-BN dose 2 receipt.**
(DOCX)

**S2 Fig. Flow chart of population included in study.**
(DOCX)

**S3 Fig. Time in days between first and second dose among individuals who had first dose before September 30, 2022 versus on/after September 30, 2022.**
(DOCX)

**S1 Table. Definitions of variables used.**
(DOCX)

**S2 Table. Descriptive characteristics of individuals who received at least 1 dose of the MVA-BN vaccine in Ontario, Canada, between June 6, 2022, to October 31, 2023, stratified by those who received dose 2 versus those who received only dose 1, and presented with row percentages.**
(DOCX)

**S3 Table. Factors associated with dose 2 receipt versus only receiving dose 1 of MVA-BN vaccine in Ontario, Canada, between June 6, 2022, to October 31, 2023, without age adjustment and with age adjustment.**
(DOCX)

## Acknowledgments

We acknowledge colleagues at Public Health Ontario for access to vaccination data from the Digital Health Information Repository, case level data from the integrated Public Health Information System, and laboratory data from LabWare. We also thank the staff of Ontario's public health units who are responsible for mpox case and contact management and data collection. We acknowledge the work of more than 30 community-based organisations across Ontario who led vaccine mobilisation and implementation with public health units and healthcare providers. This document used data adapted from the Statistics Canada Postal Code Conversion File, which is based on data licensed from Canada Post Corporation, and/or data adapted from the Ontario Ministry of Health Postal Code Conversion File, which contains data copied under license from Canada Post Corporation and Statistics Canada. Parts of this material are based on data and/or information compiled and provided by Immigration, Refugees and Citizenship Canada (IRCC) Permanent Residents Database

current to March 2023, Ontario Ministry of Health, Canadian Institute for Health Information, Statistics Canada, and IQVIA Solutions Canada. The analyses, conclusions, opinions, and statements expressed herein are solely those of the authors and do not reflect those of the funding or data sources; no endorsement is intended or should be inferred. Adapted from Statistics Canada, Canadian Census 2016. This does not constitute an endorsement by Statistics Canada of this product. We thank IQVIA Solutions Canada for use of their Drug Information File.

## Author contributions

**Conceptualization:** Ramandip Grewal, Sharmistha Mishra, Sarah A Buchan.

**Formal analysis:** Cindy Lau, Li Bai.

**Funding acquisition:** Jeffrey C Kwong.

**Methodology:** Ramandip Grewal, Cindy Lau, Jeffrey C Kwong, Ann N Burchell, Lindsay Friedman, Christine Navarro, Evaezi Okpokoro, Darrell H S Tan, Austin Zygmunt, Sharmistha Mishra, Sarah A Buchan.

**Visualization:** Ramandip Grewal.

**Writing – original draft:** Ramandip Grewal.

**Writing – review & editing:** Ramandip Grewal, Cindy Lau, Jeffrey C Kwong, Ann N Burchell, Lindsay Friedman, Christine Navarro, Evaezi Okpokoro, Darrell H S Tan, Austin Zygmunt, Li Bai, Sharmistha Mishra, Sarah A Buchan.

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
