## [Decision Letter · Decision Letter 0]

6 Jul 2025

PGPH-D-25-00546

Characteristics of individuals who received a complete, 2-dose mpox vaccine regimen as part of the public health response to the mpox epidemic in Ontario, Canada: A CIRN study

Dear Dr. Grewal,

Thank you for submitting your manuscript to PLOS Global Public Health. After careful consideration, we feel that it has merit but does not fully meet PLOS Global Public Health’s publication criteria as it currently stands. Therefore, we invite you to submit a revised version of the manuscript that addresses the points raised during the review process.

We look forward to receiving your revised manuscript.

Kind regards,

Yogesh Hooda

Academic Editor

Journal Requirements:

Additional Editor Comments (if provided):

Please carefully go through the comments from both reviewers. Please focus on Reviewer 1's comments on statistical analysis.

Reviewers' comments:

Reviewer's Responses to Questions

**Comments to the Author**

1. Does this manuscript meet PLOS Global Public Health’s publication criteria?

Reviewer #1: Yes

Reviewer #2: Yes

2. Has the statistical analysis been performed appropriately and rigorously?

Reviewer #1: No

Reviewer #2: Yes

3. Have the authors made all data underlying the findings in their manuscript fully available (please refer to the Data Availability Statement at the start of the manuscript PDF file)?

Reviewer #1: No

Reviewer #2: No

4. Is the manuscript presented in an intelligible fashion and written in standard English?

Reviewer #1: Yes

Reviewer #2: Yes

Reviewer #1: The study addresses a timely issue in the aftermath of the 2022 Mpox outbreak, examining factors associated with receiving the second dose of MVA-BN in Ontario. It identifies key barriers, such as access and healthcare engagement, which can inform interventions to improve vaccine coverage. However, it is necessary to address questions that remain unclear in the analysis and presentation of the results.

1. It is necessary to include a flowchart of population selection following the STROBE guide

2. In the study design section, it is necessary to specify that this is an observational study, indicate the type of cohort, and clarify whether it is retrospective or prospective

3. In the analysis, it is unclear why the authors used prevalence ratios (PR), which are primarily used in cross-sectional studies, to identify associations instead of relative risks (RR). Although the analyses were conducted using Poisson regression and the results are presented as RRs, this approach is not standard. Therefore, the authors should justify this choice.

4. In the analysis, it is necessary to first conduct normality tests for numerical variables, and then determine whether Table 1 presents the mean with ± SD or the median with IQR — but not both.

5. Likewise, normality analyses for numerical variables should clarify whether it is appropriate to use the Kruskal-Wallis test or the Student's t-test to compare group means.

6. The use of a standardized difference ≥0.1 as 'programmatically significant' is an arbitrary threshold commonly applied in observational studies, particularly for assessing covariate balance before and after matching or weighting procedures such as propensity score matching or inverse probability weighting. However, this threshold does not necessarily imply clinical or statistical significance relevant to the study. It is unclear whether this value was used to select variables for the adjusted analysis or to assess the success of the matching or adjustment. It is important to specify its intended purpose in the results; otherwise, it would be preferable not to mention it.

7. In the analyses, why was no variable other than age included in the adjusted model, despite the epidemiological and clinical importance of other factors? I believe that, in addition to age, sex should be included as an adjustment variable, as it is associated with both the exposure (risk behaviors) and the outcome.

8. In Table 1, percentages should be calculated based on row totals rather than column totals. This approach facilitates a clearer comparison between covariates and the event of interest, and it is also the standard practice for presenting percentages.

9. In Table 2, for the variables Neighbourhood Income Quintile and Neighbourhood Visible Minorities Quintile, missing values are excluded from the analyses; this should ideally be noted in a table footnote.

10. In Table 2, the first column (Received 2 doses) repeats the values from Table 1, but with a different percentage calculation. I suggest replacing this column with the results of a crude (unadjusted) analysis, which would help clarify whether any changes occurred in the adjusted analysis.

Reviewer #2: Thank you for the opportunity to review your manuscript, "Characteristics of individuals who received a complete, 2-dose mpox vaccine regimen as part of the public health response to the mpox epidemic in Ontario, Canada: A CIRN study."

This is a timely and important study that addresses a critical public health issue regarding mpox vaccine series completion in Ontario. The use of a large, population-based dataset is a significant strength. I offer the following comments and suggestions to help clarify and strengthen the manuscript for publication.

Major Comments

Methodological Clarity and Justification of Effect Measure: The study's design and analytical choices require further clarification.

1. Study Design: In the abstract (lines 44-49) and Methods section, the study design is not explicitly stated. While it appears to be a retrospective cohort study using secondary administrative data, this should be clearly articulated for the reader. Please specify the study design and explicitly state that this was an analysis of secondary data.

2. Choice of Prevalence Ratio (PR): The use of a prevalence ratio as the measure of association is a critical methodological point that needs robust justification. The study follows a cohort of individuals over a period (June 2022 to October 2023) to ascertain an outcome (receipt of a second vaccine dose). In such a longitudinal design, a risk ratio (RR) or incidence rate ratio (IRR) is typically the more appropriate measure of association, as it quantifies the probability or rate of the outcome occurring over time. A prevalence ratio is generally reserved for cross-sectional studies where exposure and outcome are measured at a single point in time. Please provide a detailed rationale for choosing the prevalence ratio over a risk ratio. If the analysis was conducted as a cross-sectional assessment at the end of the study period, this must be explicitly stated and justified as the optimal approach for your research question. Without this justification, the strength and interpretation of the findings are unclear.

Minor Comments

Title: The title is informative but contains an abbreviation ("CIRN") that may not be familiar to all readers. Per journal guidelines, it is best practice to avoid abbreviations in the title. Please spell out the Canadian Immunization Research Network or remove the acronym.

Abstract:

Methods (Lines 44-49): The description of the methods is vague. Please clarify the study population and how individuals were identified and included. Stating that you used "provincial health administrative data" is good, but a more explicit description of the cohort definition would be beneficial.

Results (Line 52): The phrasing "e.g., Ottawa versus Toronto" is ambiguous. Does this mean this is just one example of a comparison, or is this the specific comparison being reported? Please rephrase for clarity, for instance: "higher in Ottawa compared to Toronto..."

Conclusion (Lines 58-59): The objective was to understand how individuals who received two doses differed from those who received only one. The conclusion should more directly summarize these key differences that were identified. For example, "Individuals who completed the 2-dose series were more likely to have higher healthcare engagement, as evidenced by..."

Introduction:

Line 103: The statement regarding a sustained increase in mpox cases in 2024 is a strong justification for the study's importance. Please provide a citation to support this claim.

General:

Please ensure consistency in the style of in-text citations throughout the manuscript to conform to the journal's formatting requirements.

I trust these comments will be useful in revising your manuscript. I look forward to seeing a revised version that addresses these points.

**Do you want your identity to be public for this peer review?** For information about this choice, including consent withdrawal, please see our Privacy Policy

Reviewer #1: **Yes: ** Hugo Arroyo-Hernández

Reviewer #2: **Yes: ** Suleiman Idris Ahmad

---

## [Decision Letter · Decision Letter 1]

28 Oct 2025

Characteristics of individuals who received a complete, 2-dose mpox vaccine regimen as part of the public health response to the mpox epidemic in Ontario, Canada

PGPH-D-25-00546R1

Dear Dr Grewal,

We are pleased to inform you that your manuscript 'Characteristics of individuals who received a complete, 2-dose mpox vaccine regimen as part of the public health response to the mpox epidemic in Ontario, Canada' has been provisionally accepted for publication in PLOS Global Public Health.

Best regards,

Yogesh Hooda

Academic Editor

Reviewer Comments (if any, and for reference):

Reviewer's Responses to Questions

**Comments to the Author**

Reviewer #1: All comments have been addressed

publication criteria?

Reviewer #1: Yes

3. Has the statistical analysis been performed appropriately and rigorously?

Reviewer #1: Yes

4. Have the authors made all data underlying the findings in their manuscript fully available (please refer to the Data Availability Statement at the start of the manuscript PDF file)?

Reviewer #1: No

5. Is the manuscript presented in an intelligible fashion and written in standard English?

Reviewer #1: Yes

Reviewer #1: (No Response)

**Do you want your identity to be public for this peer review?** For information about this choice, including consent withdrawal, please see our Privacy Policy

Reviewer #1: **Yes: ** Hugo Arroyo-Hernández
